cybernetics/robotics/human-computer interaction

anticipating synchronization, coupled oscillators, synchronization, time delay, collaborative robotics

**Author for correspondence:**
Henry Eberle
e-mail: h.eberle@ucl.ac.uk

# Synchronization-based control for a collaborative robot

Henry Eberle[1], Slawomir J. Nasuto[2] and Yoshikatsu Hayashi[2]

[1]Department of Orthopaedics and Musculoskeletal Science, Division of Surgery, University College London, London WC1E 6BT, UK
[2]Brain Embodiment Lab, Biomedical Engineering, School of Biological Sciences, University of Reading, Reading RG6 6AH, UK

(iD) HE, 0000-0002-1507-2643

This article introduces a new control scheme for controlling a robotic manipulator in a collaborative task, allowing it to respond proactively to its partner's movements. Unlike conventional robotic systems, humans can operate in an unstructured, dynamic environment due to their ability to anticipate changes before they occur and react accordingly. Recreating this artificially by using a forward model would lead to the huge computational task of simulating a world full of complex nonlinear dynamics and autonomous human agents. In this study, a controller based on anticipating synchronization, where a 'leader' dynamical system is predicted by a coupled 'follower' with delayed self-feedback, is used to modify a robot's dynamical behaviour to follow that of a series of leaky integrators and harmonic oscillators. This allows the robot (follower) to be coupled with a collaborative partner (leader) to anticipate its movements, without a complete model of its behaviour. This is tested by tasking a simulated Baxter robot with performing a collaborative manual coordination task with an autonomous partner under a range of feedback delay conditions, confirming its ability to anticipate using oscillators instead of a detailed forward model.

## 1. Introduction

To create a true collaborative robot, one that can physically cooperate in a task, the robot must be able to anticipate its partner's behaviour. This can be clearly understood from human-to-human interactions such as manipulating a large object or even dancing; any delay in reacting to the partner may cause a breakdown in the collaborative movement and a failure of the shared task. It is particularly important that a robot provides a seamless cooperative experience because it cannot

communicate with its partner to improve its performance. Thus, conventional (that is to say, reactive) closed-loop control is not the ideal paradigm for controlling a collaborative robot.

A common approach is for collaborative robots to attempt to infer a human partner's intentions from their behaviour so that the robot can 'pre-empt' their future actions. However, while many stereotypical human movements are well characterized (such as simple reaching motions, by the minimum-jerk model [1]), there is no accepted complete model of human behaviour. Various methods have been used to work around this limitation, such as in the study by Thobbi et al., where the robot switches to a reactive follower role when its prediction of the human's motion diverges from its sensor measurements [2]. Mainprice et al. attempted to learn the objective function of human reaching movements through inverse reinforcement learning, taking the assumption that humans operate on the principle of optimal control [3], while Bussy et al. decomposed their collaborative task into motor primitives with rigidly defined trajectories [4].

These studies assume a dichotomy between fully understanding a partner's intentions and merely reacting to their actions, but it is also possible to consider how to enable the robot to behave proactively in the short term, maximizing its ability to respond to unexpected movements. Although attempting to anticipate an imperfectly understood partner might seem counterintuitive, the study by Ishida and Sawada [5] suggests that it in fact allows the minimization of transient errors when the target moves abruptly. This comes at the cost of slightly lesser accuracy during steady motion, but it still gives superior performance over a reactive controller. A good framework for approaching this problem is found in strong anticipation [6], which relates to the idea that organisms like human beings anticipate events by continuously coupling their own (cognitive or somatic) processes to the dynamics of the environment, contrasting with methods such as MOSAIC [7] that need to construct a model of the predicted system ('weak' anticipation). The most applicable form of strong anticipation is anticipating synchronization (AS), where a 'follower' dynamical system synchronizes with the future of a 'leader', rather than its instantaneous state [8]. This is not a violation of causality, but relies on the fact that a deterministic dynamical system's current state is strongly determined by its past. In the most commonly used version of AS, the difference between the state of the leader and the delayed state of the follower constitutes a coupling term ($K[x(t) - y(t - \tau)]$) that will drive the follower towards the future of the leader

$$\dot{x}(t) = f_1(x(t)) \tag{1.1}$$

and

$$\dot{y}(t) = f_2(y(t)) + K[x(t) - y(t - \tau)]. \tag{1.2}$$

If the dynamics of the leader ($f_1(x)$) and follower ($f_2(y)$) are similar, and the delay term ($\tau$) is not too large, the joint system will reach a stable state where $y(t) = x(t + \tau)$ and the coupling term disappears [9]. Because many dynamical systems exhibit similar behaviour over short timescales, $f_1$ does not have to equal $f_2$, and the follower can be governed by significantly simpler dynamics than the leader. Thus, there is no need to model the higher level decision-making processes of a partner, so long as a follower system can be designed that can anticipate the short-term dynamics of their motor behaviour.

AS has typically been applied to isolated pairs or chains [10] of autonomous systems since its discovery, but there have been studies that attempt to combine it with feedback control. An AS-based predictor was used by Oguchi and Nijmeijer to replace a standard predictive model in a system that would otherwise be too nonlinear for most predictive control methods [11]. More recently, a similar method was used to enable the control of mobile robots via long-distance telecommunication [12] without experiencing lag.

These past examples stick as closely as possible to existing predictive control methods, but AS allows a new relationship with delay. If the follower is considered to be the plant in a control sense, then the delayed self-feedback becomes a measurement of the plant's state, and the delay term becomes the intrinsic lag between a control signal and the plant's measured response. In this framework, any additional delay, either due to signal transmission time or response lag in the plant, will be counteracted by a strengthening of the coupling term, causing anticipatory behaviour.

Obviously this behaviour is not normally exhibited by closed-loop controllers, which can be easily explained by the fact that the controller (leader) and plant (follower) do not typically have similar dynamical behaviour. Exploiting the unique delay response of AS therefore requires a different form of control law. In [13], a framework was developed for a 'parallel' controller that uses an internal model of the robot to enforce a specific dynamical behaviour on the plant, allowing it to be coupled with a set of dissimilar target dynamics in an AS leader/follower relationship. Reproducing the dynamics of the robot in the model is not necessarily optimal, however, as they may be very different to those of the leader. Therefore, in this study, a new set of follower dynamics was designed, where the joint and motor

dynamics of the robot are replaced with simpler systems that have a proven capability to anticipate much more complex leaders: leaky integrators and harmonic oscillators. Despite their simplicity, leaky integrators, dynamical systems that 'relax' to a fixed resting state, can anticipate a variety of phenomena through the effect of negative group delay [14,15]. This is where certain frequency components of a signal are propagated as if by an (imagined) filter with negative delay. Conversely, harmonic oscillators have shown a surprising ability to anticipate 'unpredictable' chaotic dynamics with similar local features [6] and exhibit self-organizing behaviour when coupled together [16,17] that indicates they may be ideal components for a joint system such as a model of a robot arm.

Rather than creating a special-purpose robot that could call into question the broader applicability of this study, the Baxter collaborative robot was chosen as a testing model. Baxter's hardware and software safety features provide a reasonable estimate of the constraints, which a collaborative control rule can be expected to have, and it has a high-quality simulator that can be used for repeated experiments [18]. The experimental can be tailored to the intrinsic timescale and capabilities of the Baxter platform: a back-and-forth coordinated motion in which the robot must follow a moving target representing its partner without lagging or becoming unstable. This simulates the timing aspect of bimanual handling (e.g. moving one end of a piece of furniture) without mutual physical interaction that would require a specialized platform to investigate.

This study aims to demonstrate that a control system designed from the bottom up to take advantage of AS can enable a robot to anticipate the motion of an autonomous partner, even without an accurate model. By tying the velocities of its joints to a series of simulated dynamical systems, the robot's own delayed sensory feedback becomes the key element in an AS coupling that drives it to follow its partner with minimal (or even slightly negative) latency. Using the accurate Gazebo simulation of the Baxter robot allows repeated experiments over a large range of parameters to confirm the robustness of the controller. This represents a categorically different approach from methods of collaborative control based on explicit models, which decline in performance if the partner is not correctly characterized.

# 2. Methods

## 2.1. Robotic platform and simulation

The Baxter robot consists of two 7-degree-of-freedom arms mounted on either side of a central column capped with an articulated screen 'head', together constituting a humanoid 'torso' [19]. This is attached to a 0.91 m pedestal, giving the robot a total height of 1.85 m. The arm joints are referenced with alphanumeric labels from $S_1$ at the base to $W_1$ at the wrist, as shown in figure 1. As a collaborative robot, Baxter is primarily designed for safety when interacting with humans, which is achieved through a combination of specialized hardware and software. Each arm joint contains a series elastic actuator [21] that allows control of joint stiffness and direct measurement of joint torque. Combined with a real-time onboard controller, this allows the robot to maintain arm velocities and forces that do not risk serious harm to the humans around it. A secondary effect of these safety constraints is that Baxter will not accept commands that cause self-collision (defined as any element of its kinematic model intersecting any other).

In designing the task used to test the new control scheme, a conscious effort was made to remain within some of these constraints by utilizing Baxter's velocity control facility. Unlike the default position control, this allows the velocity at each joint to be set directly, but still prevents them exceeding a maximum value of $2 \, \text{rad s}^{-1}$ (or $4 \, \text{rad s}^{-1}$ for the three terminal wrist joints) and automatically halts the robot in the event of a collision or a sustained 'crush' against an external object. The ability to control the robot's velocity this way is crucial to implementing AS as it depends on altering the time evolution of the follower, so this was considered the best compromise between safety and the requirements of the study.

During the testing phase, to evaluate the control under an appropriate range of conditions within a reasonable time and without risking wear-and-tear to the robot's components, the motion of the robot can be simulated by an accurate dynamical model, simulated in Gazebo [18] (see figure 2). This replicates all of Baxter's software interfaces as well as its characteristic dynamical behaviour, allowing the results of the simulated experiments to be generalized to the physical robot [22].

## 2.2. Collaborative task

The aim of this study is to create a controller suitable for proactive cooperation with a partner, so a coordinated task was defined where the robot must match a partner's motion, replicating its trajectory

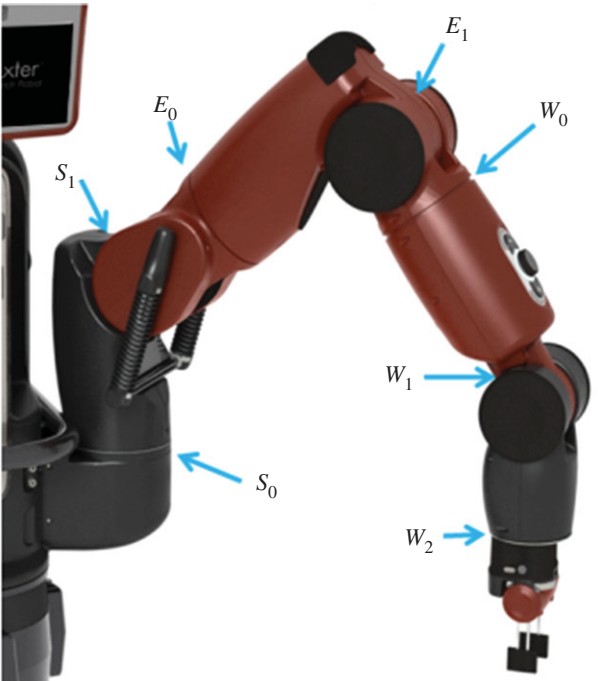

**Figure 1.** Joint positions on Baxter's left arm, split into shoulder joints $S_0$ and $S_1$, elbow joints $E_0$ and $E_1$, and wrist joints $W_0$, $W_1$ and $W_2$. All right arm joints are identical to those on the left, except that $S_0$, $E_0$, $W_0$ and $W_2$ are mirrored. Image from [20].

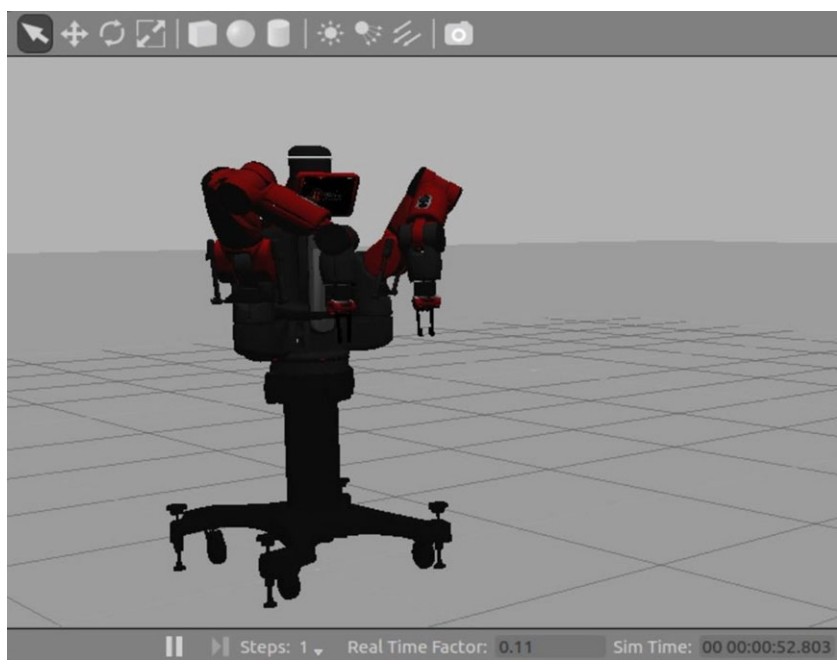

**Figure 2.** Gazebo simulation of the Baxter robot in action-ready 'untucked' pose. The simulation was ordinarily run without the GUI to speed up processing, but the graphical representation was used for manual error checking.

along the X-axis as shown in figure 3. To eliminate lag, the robot must not merely follow its partner, but actively anticipate their movements. The robot is considered to be the follower in all these actions, in line with expected use cases for a collaborative robot.

An autonomous partner that provides unpredictable behaviour to the robot is represented by a periodic oscillator with pseudo-random variation in its amplitude. This is not intended to be a reproduction of a true human movement, but to simulate motion that has consistent local features

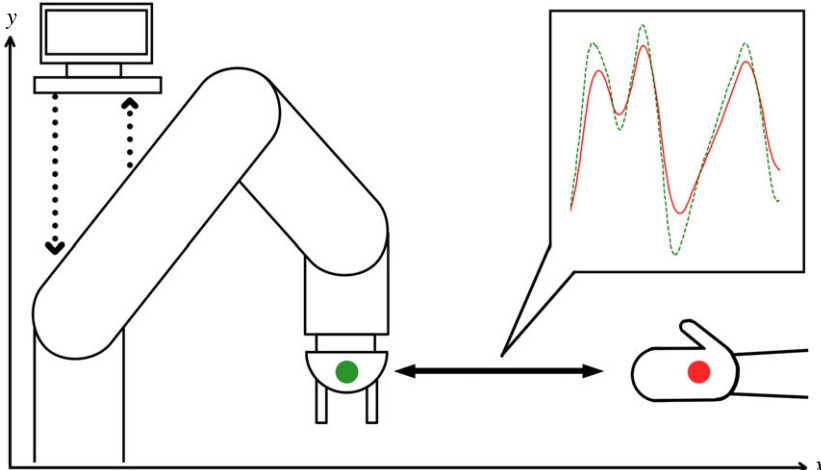

**Figure 3.** Illustration of simulated manual coordination task: the robot (green, follower) is controlled by the AS control system and must match the movements of the leader (red) as closely as possible, despite its pseudo-random motion.

(sinusoidal peaks), but cannot be exactly predicted using a forward model (assuming no access to an identically seeded pseudo-random number generator).

## 2.3. Parallel AS controller

Anticipation requires the follower to have autonomous behaviour that is sufficiently similar to the leader's, and preliminary results suggested that despite its inbuilt elasticity at the joints, Baxter's unmodified dynamics did not meet this requirement. In the 'parallel system' of [13], stable anticipation is enabled by using an internal model of a non-delayed control loop to impose an 'idealized' dynamical behaviour on the robot, allowing it to anticipate a target with the addition of an AS coupling term. In that study, the internal feedback loop identically reproduced the dynamics of the robot, but it was not proven that this was an optimal solution for producing AS in robotic control. For this study, the dynamics of the internal feedback loop were instead modified to represent systems with a proven capability to anticipate a range of complex systems: leaky integrators [14] and harmonic oscillators [6]. This was achieved by creating a simulated control loop where a 'robot' with identical kinematics to Baxter, but actuator dynamics replaced with one of these new systems, tracks the target without feedback delay. The output of this internal feedback loop reproduces the same behaviour in the real robot's joints, allowing it to act as an effective AS follower when coupled.

As shown in figure 4, the velocity command to the robot $v(t)$ is a linear combination of the time derivative of the follower system $\dot{q}_{\text{follower}}(t)$ and the coupling term $u_{\text{couple}}$. Expanding $u_{\text{couple}}$ gives the full control equation for the robot

$$v(t) = B\dot{q}_{\text{follower}}(t) + KM[x_{\text{follower}}(t) - x_{\text{plant}}(t - \tau)], \qquad (2.1)$$

where $v$ is the velocity command to the robot's actuators, $\dot{q}_{\text{follower}}(t)$ is the time derivative of the follower model's state and $x_{\text{plant}}(t - \tau)$ and $x_{\text{follower}}$ are the positions of the robot's end effector and the model's 'end effector' (calculated by using a copy of the robot's forward kinematics, $s(.)$), respectively. $B$ and $K$ are scalar constants representing the relative contributions of the follower model and the coupling term, and $\tau$ is a feedback delay term. The positions of the modelled ($x_{\text{follower}}$) and actual ($x_{\text{plant}}$) robot end-effectors are expressed in spherical coordinates (radius ($r$), polar angle ($\theta$) and azimuthal angle ($\phi$)). This allows a simple transformation, $M$, to map the individual spherical coordinates onto the joints $S_0$, $S_1$ and $E_1$, with all other joints fixed via an independent high-gain control loop. $\theta$ and $\phi$ are controlled by joints $S_0$ and $S_1$, respectively, while $r$ can be controlled by using joint $E_1$ to extend the arm:

$$M = \begin{bmatrix} 0 & 0 & 1 \\ 0 & 1 & 0 \\ -1 & 0 & 0 \end{bmatrix}. \qquad (2.2)$$

Because $S_1$ also affects the radial coordinate, $M$ is not a true coordinate transformation; however, since the joints are effectively coupled through the forward kinematics, $E_1$ can continuously compensate for $S_1$'s effects on $r$. This limits the arm to variations on a single pose, allowing the static linear transformation

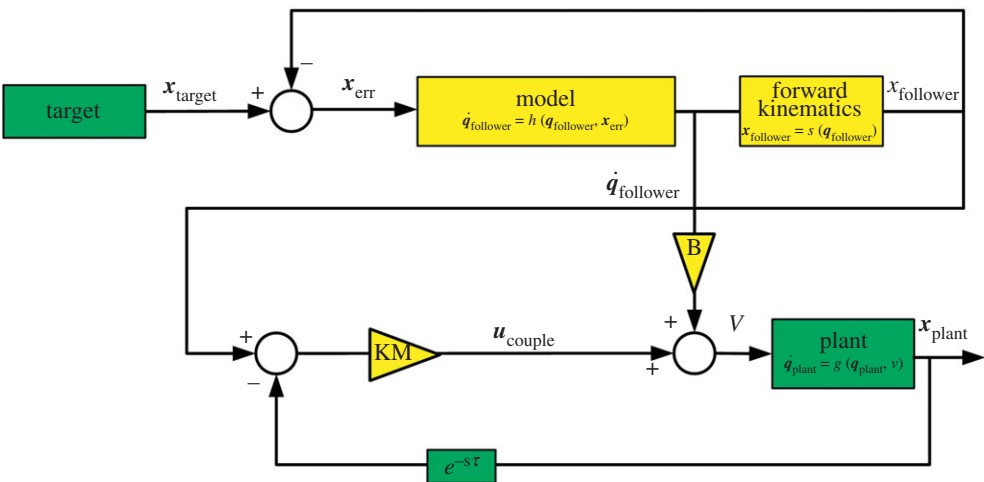

**Figure 4.** Block diagram of the proposed control scheme, where the velocity command $v$ to the plant (Baxter) is a linear combination of a velocity control signal $B\dot{q}_{follower}$ produced by an internal control loop with specific 'follower' dynamics, and a second 'coupling' control signal $u_{couple} = KM[x_{follower}(t) - x_{plant}(t - \tau)]$ based on delayed feedback from the robot itself, where $B$ and $K$ are scalar constants and $M$ is a transformation matrix. The follower model $h(q_{follower}, x_{err})$ tracks the position of the target $x_{target}$ by minimizing the difference $x_{err}$ between $x_{target}$ and the model's 'end effector' calculated using a copy of Baxter's forward kinematics ($x_{follower} = s(q_{follower})$). The internal control loop imposes a specific dynamical behaviour on the plant, causing it to act as an AS follower that can anticipate the target's movements. Thus, the addition of the coupling term causes the robot to lead the target if the feedback delay ($\tau$) is large enough. (Green blocks represent intrinsic elements of the underlying system, while yellow blocks have been added to enable anticipation.)

to ensure convergence on the target. This is preferable to using inverse kinematics because, in particular, the use of a Jacobian-based method becomes problematic when calculating the difference between two different poses separated by a delay as in the coupling term.

The follower's prediction occurs via a simple variation on dynamical synchronization. While an ordinary synchronization coupling ($[x(t) - y(t)]$) represents a difference term that drives the follower $y$ to evolve towards the current state of the leader $x$ ($[x(t) = y(t)]$), a delay coupling ($[x(t) - y(t - \tau)]$) approximates a term that accelerates the follower's time evolution towards the leader's future state ($[x(t + \tau) = y(t)]$), assuming they are sufficiently similar. The study by Hayashi *et al.* [9] revealed that this is due to the renormalization of time in the follower: when $y(t - \tau)$ from equation (1.2) is Taylor expanded, it gives the new time step $t^* = t/(1 - k\tau)$, causing the follower to evolve faster than the leader until it 'catches up' with its future state. Provided a transversal system $\dot{\Delta} = f_1(x) - f_2(y) - K\Delta$ can be defined (where $\Delta = x - y$) that has a fixed point at $\Delta = 0$ (which is a trivial solution), there exists some $\tau < \tau_0$ for which that fixed point is locally attracting [10]. The size of this region cannot be guaranteed and can be estimated only numerically for any pair of systems, which motivated the selection of follower dynamics that were established from prior research to have a significant anticipation period for more complex leaders. In this study, even though the autonomous dynamics of the follower system were replaced with leaky integrators and harmonic oscillators, the parallel structure of the control scheme (figure 4) reinforces this internal model to induce resonance with the dynamics of the leader system, resulting in the creation of similar dynamics.

### 2.3.1. Leaky integrator follower

One of the simplest forms of dynamical system, a leaky integrator, is an integrator that gradually evolves towards a fixed resting state when unperturbed. The justification for using an internal model based on leaky integrators comes from prior studies on the phenomenon of negative group delay [14,15], where it was shown that such a system can predict a low-frequency leader up to a theoretical maximum of half the coupling delay ($\tau/2$ s). Although a leaky integrator follower cannot perfectly predict more complex systems, it can reliably anticipate a range of frequency components that, for some signals, can represent the majority of the salient information. In particular, negative group delay

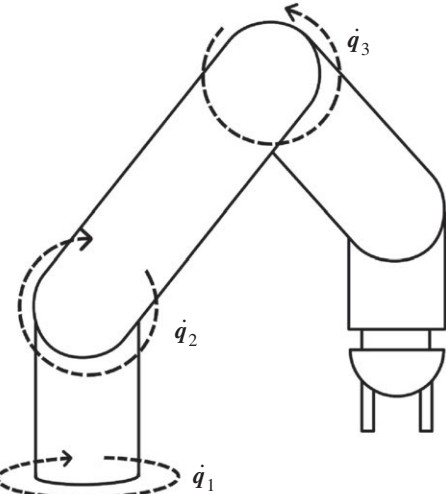

**Figure 5.** Illustration of the internal model used within the parallel controller. The kinematics of the model are the same as the actual robot, but the velocities at the joints are replaced with $\dot{q}_{\mathrm{follower}}$.

has been posited as an explanation for human anticipatory movements of the kind this experiment attempts to mimic.

Equation (2.3) describes the form of the leaky integrator used in this study

$$\dot{q}_{\mathrm{follower}}(t) = -Aq_{\mathrm{follower}}(t) + GM[x_T(t) - s(q_{\mathrm{follower}}(t))], \tag{2.3}$$

where $A$ is a constant diagonal matrix of a time constant $a$ ($A = aI$), $G$ is a gain constant, $x_T$ is the position of the target and $s(.)$ represents the forward kinematics of the robot. $x_T$ is a vector of resting states corresponding to the centre of each joint's range of motion. The drive term $GM[x_T(t) - s(q_{\mathrm{follower}}(t))]$ causes the 'end effector' of the follower calculated with the forward kinematics $s(q_{\mathrm{follower}})$ to track the target, making the model behave like a copy of the robot with joint dynamics governed by equation (2.3) (as illustrated in figure 5).

### 2.3.2. Harmonic oscillator follower

Harmonic oscillator followers are known to be successful at anticipating systems that exhibit smooth, sinusoidal peaks [6], even if the overall system behaviour is chaotic or otherwise unpredictable, as in this case. In addition, networks of coupled oscillators are known to exhibit self-organizing behaviour [16,17], which is necessary for a group of dynamical systems to collectively anticipate a single leader

and
$$\left.\begin{array}{l} \dot{q}_{\mathrm{follower}}(t) = -An(t) + GM[x_T(t) - s(q_{\mathrm{follower}}(t))] \\ \dot{n}(t) = Aq_{\mathrm{follower}}(t) - GM[x_T(t) - s(q_{\mathrm{follower}}(t))] \end{array}\right\}. \tag{2.4}$$

All terms are the same as in the leaky integrator system, except for the term $n$, which tracks the oscillators' state and feeds back a negative copy that triggers sinusoidal oscillations with a frequency proportional to the constant $A$.

## 2.4. Robot operating system implementation

The architecture of a robot operating system (ROS) consists of multiple independent software 'nodes' that communicate asynchronously by publishing and subscribing to a series of 'topics', allowing information to be shared without the nodes being aware of each others' existence [23]. It should be noted here that 'asynchronous' refers only to the mode of node-to-node communication and has no relation to the dynamical synchronization being explored in this study.

The ROS implementation of the experiment was composed of three elements: the Baxter robot simulation and two software 'nodes' running on a Linux workstation connected via a local network. The target (or leader) node publishes a pseudo-random waveform that is subscribed to by a second control (or follower) node. This publishes a control signal that is subscribed to by the robot, and

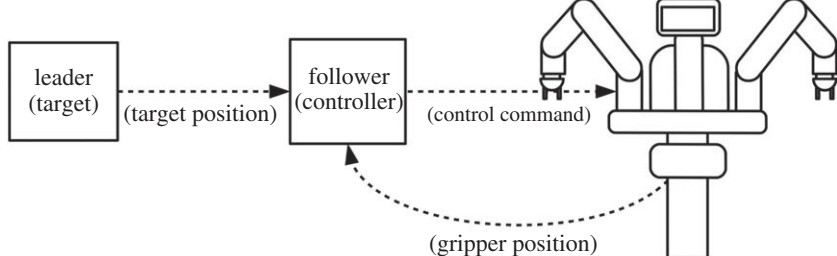

**Figure 6.** Diagram of the relationship between the Baxter robot (or Gazebo simulation thereof) and the ROS nodes controlling its behaviour. A velocity control command is produced by the follower node based on the difference between the target and the robot's gripper position.

subscribes in turn to the position of the robot's end effector, as calculated by its internal PC based on measurements from encoders at each joint. The relationship between these elements is illustrated in figure 6. The Gazebo simulation of the Baxter robot [18] replicates all of its software interfaces, in addition to it kinematics and dynamics, and thus can be inserted in place of the real robot without altering any other element of the system.

Because of ROS's asynchronous implementation, the exact relationship between the timings of the leader and follower nodes cannot be guaranteed. For this reason, a buffer was implemented within the follower node that delays and stores the feedback from the robot for a number of (0.01 s) iterations. For each iteration, the node publishes the leader, follower and robot feedback values used to update the follower's state, which are recorded as experimental data. This means that the additional feedback delay remains consistent with respect to the follower system for any given time step, and the output of the system is compared to the input that caused it.

## 2.5. Testing

To test this control method, the Gazebo simulation of the Baxter robot attempts to follow a partner that oscillates along the $X$-axis with a constant frequency of 0.5 Hz, but pseudo-random amplitude (generated by applying a moving average filter to the output of NumPy's 'random()' function). Both candidate follower systems are tested over a range of feedback delay ($\tau = 0.01$–0.6 s) and coupling strength ($K = 0$–30) and compared against the performance of a basic proportional controller that uses the same transformation matrix (equation (2.5)).

$$v(t) = KM[x_T(t) - x_R(t - \tau)]. \tag{2.5}$$

The degree of anticipation (or lag) is measured by taking the $X$-axis motion of the target and the robot end effector over 10 min, removing the DC offset from both and cross-correlating them, extracting the lag with the highest correlation coefficient down to a resolution of 0.01 s. All analysis is performed using Cartesian coordinates as this is the format in which Baxter natively publishes its position data.

## 2.6. Stability analysis

As explained by Voss [8], the stability of an AS system of the form in equation (1.2) is dependent on the transversal system $\Delta = x(t) - y(t - \tau)$ having a stable fixed point at $\Delta = 0$ (the anticipating manifold). For pairs of identical linear systems, a sufficient condition for the manifold's stability can be determined by calculating its complex Lyapunov exponent [9], yielding an explicit dependence on delay $\tau$ and coupling strength $K$. Such an analytical solution does not exist for nonlinear or non-identical pairs, but the stability of the manifold with respect to $\tau$ and $K$ can be determined numerically by constructing a phase diagram. By systematically varying $\tau$ and $K$ over a series of simulations, it is possible to visualize the stability of the anticipation manifold in the $K$–$\tau$ plane. As in [9], we use the correlation coefficient and correlation lag to identify the region in the $K$–$\tau$ plane in which the anticipation manifold is stable, as stable regions share the characteristics of negative lag (the follower leads the leader) and high correlation coefficient (the follower closely matches the leader). A simplified visualization of one of these phase diagrams is shown in figure 7, showing a

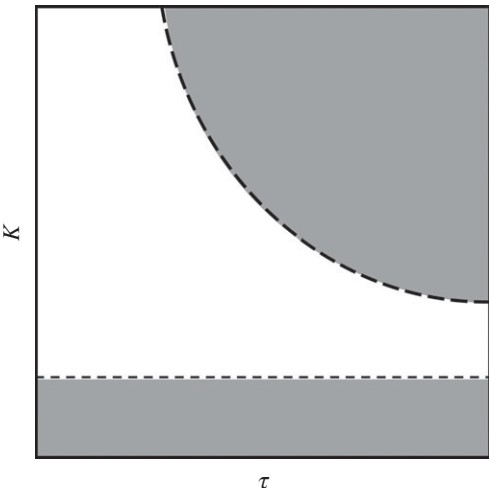

**Figure 7.** Simplified visualization of a phase diagram for the parallel AS controller described in this article, showing the stability of the anticipating manifold on the $K$–$\tau$ plane. White areas are regions in which the anticipating manifold is stable, and the robot predicts its partner, while grey areas indicate that the manifold is unstable and prediction cannot occur.

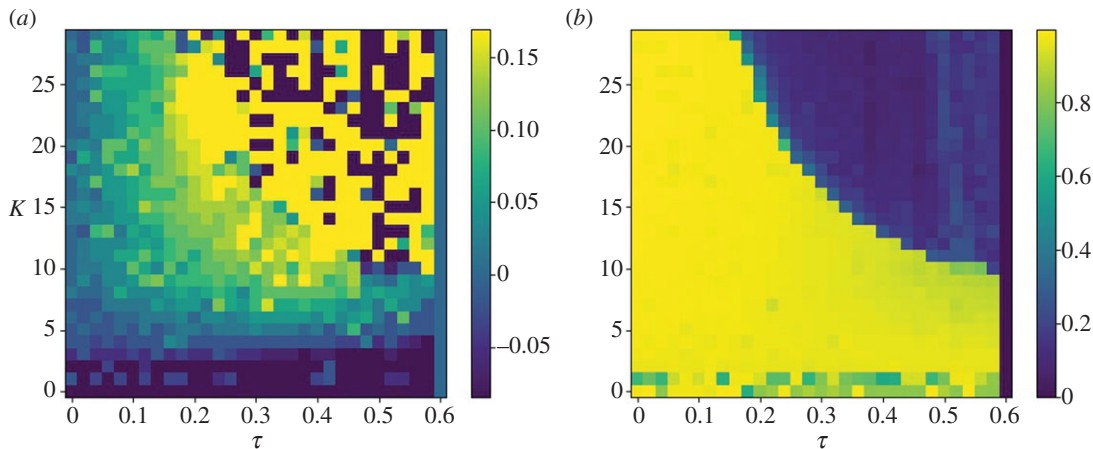

**Figure 8.** Cross-correlation peak lag (*a*) and peak correlation coefficient (*b*) plotted against coupling strength $K$ and delay $\tau$, where the dynamics of the internal model are governed by equation (2.3). Where $K > 5$, and barring the unstable region bounded by $K\tau = 6$, where the correlation coefficient is close to 0, the robot leads the target, as indicated by positive cross-correlation lag values.

characteristic division between stable and unstable regions. In the results below, the phase diagrams are presented as heatmaps to show the variation in prediction time within the stable region.

# 3. Results

## 3.1. Leaky integrator follower

Observing the cross-correlation function between target and robot depicted in figure 8*a*, it can be seen that the lead time (represented by positive values) increases over the range of $0 < \tau < 0.2$ and $K > 5$. The region of stable anticipation is bounded by the region below $K = 5$, where the robot lags the target, and the region bounded by $K\tau = 6$, where the anticipation manifold becomes unstable, as shown in figure 7. Both of these features are consistent with the approximated stability conditions described in [24]: if $K$ is not sufficiently large, the follower will continue to lag behind the leader, and if $K\tau > a$, where $a$ is a constant value determined by the nature of the leader and follower, the joint system becomes unstable and no anticipation can occur.

The lag at 0.01 s feedback delay is 0.01 s, as displayed in figure 9*a*, considerably lower than that achieved by the proportional controller in the same minimum-delay condition (0.46 s, as shown in

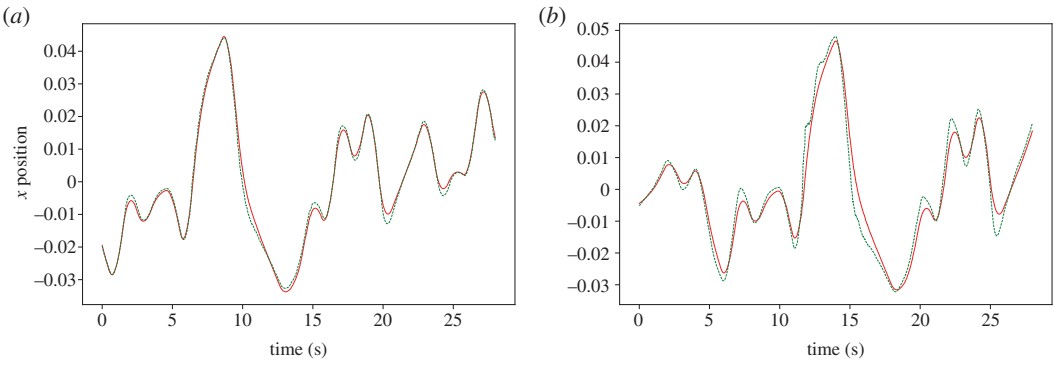

**Figure 9.** An example of the parallel controller with leaky integrator follower dynamics tracking a pseudo-random oscillating target along the X-axis, with 0.01 s (*a*) and 0.2 s (*b*) feedback delay. The target is represented by a solid line, while the robot's end effector is shown as a dashed line. An increase in feedback delay causes the robot to lead the target on average at the cost of reduced accuracy.

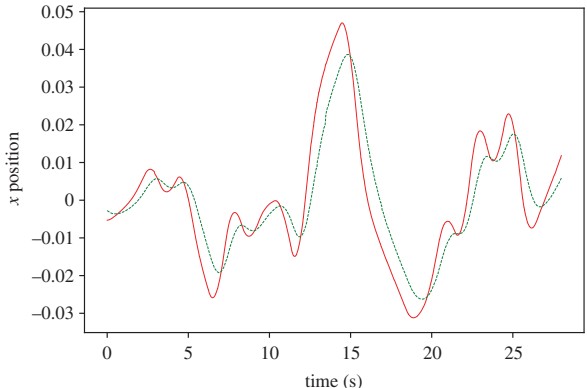

**Figure 10.** Proportional controller tracking a pseudo-random oscillating target along the X-axis, with 0.01 s feedback delay. The target is represented by a solid line, while the robot's end effector is shown as a dashed line.

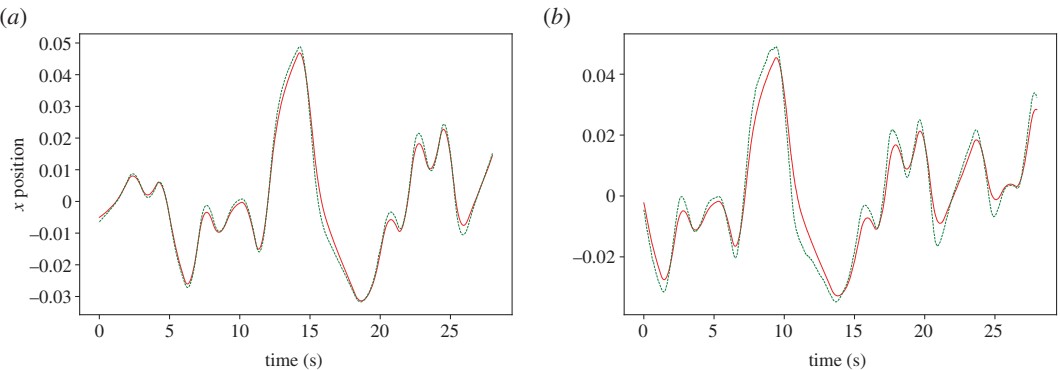

**Figure 11.** An example of the parallel controller with harmonic oscillator follower dynamics tracking a pseudo-random oscillating target along the X-axis, with 0.01 s (*a*) and 0.2 s (*b*) feedback delay. The target is represented by a solid line, while the robot's end effector is shown as a dashed line. Where feedback is delayed by 0.2 s, the robot noticeably leads the target.

figure 10) despite optimal gain. Once $\tau$ is raised to 0.2 s, the robot begins leading the target by 0.09 s, albeit with reduced fidelity to the original waveform (figure 9*b*), reflecting the decrease in cross-correlation coefficient near the stability border shown in figure 8*b*.

## 3.2. Harmonic oscillator follower

The harmonic oscillator-driven follower actually slightly leads the target by 0.03 s at 0.01 s feedback delay (figure 11*a*), but exhibits the same time response as the leaky integrator by $\tau = 0.2$ s, as reflected in the plot

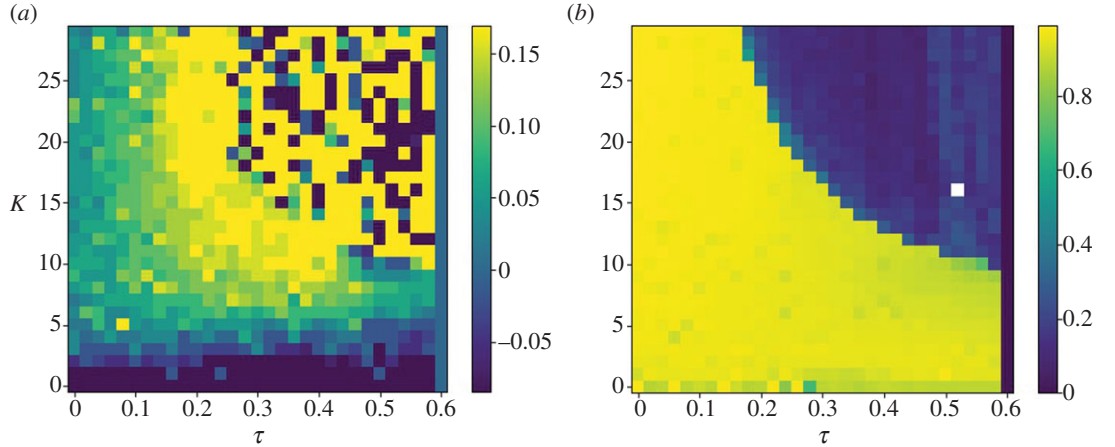

**Figure 12.** Cross-correlation peak lag (*a*) and peak correlation coefficient (*b*) plotted against coupling strength *K* and delay $\tau$, where the dynamics of the internal model are governed by equation (2.4). Where $K > 5$, and barring the unstable region bounded by $K\tau = 6$, where the correlation coefficient is close to 0, the robot leads the target, as indicated by positive cross-correlation lag values.

of the cross-correlation over $K$ and $\tau$ (figure 12). Figure 12 indicates that the robot anticipates the target throughout the range of $\tau$, except when $K$ is not sufficiently large, and within the unstable region in the top right. However, the harmonic oscillator produces fewer erroneous oscillations when anticipating nonlinear features of the target motion (compare the 5–10 s period in figures 9*b* and 11*b*).

## 4. Discussion

In the previous sections, two potential internal models were tested to stand in for the robot's dynamical behaviour, enabling a Baxter robot to act as a predictive follower. Despite being based on very simple dynamics, in both the harmonic oscillator and relaxation system cases, there was a characteristic correspondence among feedback delay, coupling strength and anticipation, consistent with Stepp's observations in [25] and the analytical predictions of the study by Hayashi *et al.* [24]. However, due to the mismatch between the (complex) leader and (simple) follower, at no point did the robot's lead time equal or exceed the feedback delay. In practical terms, this means that any control system based on this controller would exhibit minimal lag, but would not appear to anticipate the target to an outside observer. Nonetheless, this result matches the phenomenon seen in Stepp's human studies, where the delayed cursor actually perceived by the subjects never overtook the target—similarly, the robot is only 'aware' of the delayed feedback, and it is this delayed version of the end effector that converges on the target.

This behaviour is qualitatively different from that of systems where a forward model is placed in serial with the target and the plant such as Oguchi and Nijmeijer's [11]. In these, the prediction horizon is fixed, and any additional delay will 'subtract' from the prediction time. In contrast, the parallel system presented here is inherently adaptive: any increase in delay will be counteracted by an increase in anticipation, allowing the robot to operate without certainty of the delay value. This is a true step forward from the previous work in [13], showing that the same properties of robust anticipation can be applied to realistic robotic systems, even without a detailed forward model.

Of the two followers the harmonic oscillator-based follower exhibited a greater average anticipation period, unambiguously leading the target even in the lowest delay condition. This is in one way unsurprising, because the leader signal resembles an oscillation at the same natural frequency as the follower, with only the amplitude pseudo-randomized. In another respect, this is curious because the target motion is generated by low-pass filtered noise—i.e. a leaky integrator. Since the leader is driven by a pseudo-random process, neither follower can be considered a perfect match, but it is clear that the oscillators' local behaviour (sinusoid-like peaks) is sufficiently similar to the leader to produce a prediction comparable to that of the leaky integrator's.

Finally, although adding feedback delay to the system validated its ability to anticipate a target, the system's behaviour when the delay is minimal is also worth examining. Both the oscillator and relaxation dynamics successfully eliminated the lag displayed by the proportional controller under the same

conditions. This strongly indicates that anticipation is also occurring in response to the delay already present in the robot control loop, increasing the overall responsiveness of the system.

# 5. Conclusion

Although AS has properties that could be usefully applied to collaborative control, where predicting a partner's movements is crucial, it relies on a similarity in dynamics between robot (follower) and partner (leader) that simply may not exist. This study has resolved this issue by expanding upon the parallel system first demonstrated in [13]. What would ordinarily be a single control loop is split into two: the first from a forward model responsible for setting the robot's behaviour as a dynamical 'follower' and the second an AS coupling term, driving the robot to anticipate the target. The resulting controller can counteract any feedback delay that the system experiences with a proportional degree of anticipation, without requiring exact knowledge of the delay value. Our use of both leaky integrators and harmonic oscillators as the basis for the forward model to predict a pseudo-random oscillation firmly establishes that this parallel structure relaxes any requirement that the model should exactly match the robot's dynamics. In its stead is the principle that the model should represent the dynamics the robot would need to anticipate the target—a much easier condition to meet.

The results shown here demonstrate that an existing collaborative robot can display the same adaptive anticipation capability observed in human delayed-feedback tracking experiments, which is an optimal behaviour for reducing lag and transient error.

This method cannot necessarily replace the need to infer intention when a partner switches tasks, or disengages, but provides a framework for smoothly performing the type of continuous spatial interaction that is necessary for harmonious cooperation. In such scenarios, the type of adaptive anticipation exhibited here is optimal for reducing lag and transient errors that drag down performance.

Data accessibility. Data and relevant code for this research work are stored in GitHub: https://github.com/hefebe/baxter_synchronisation_citable and have been archived within the Zenodo repository: https://www.doi.org/10.5281/zenodo.4290079.

Authors' contributions. H.E. designed and ran the simulations and drafted the manuscript, as well as designing and running the experiments. S.N. and Y.H. assisted in drafting the manuscript.

Competing interests. We declare we have no competing interests.

Funding. This study was partially funded by the University of Reading faculty funding for PhD studentship.

Acknowledgements. Thanks to Christopher Buckley, for encouraging me to finally get my work to publishing point, and everyone at CREATe for providing me with the environment to finally do so.

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
