## [Reviewer comments · Royal Society Open Science]

Review History

RSOS-191762.R0 (Original submission)

Review form: Reviewer 1 (Meghan Huber)

Is the manuscript scientifically sound in its present form?

No

Are the interpretations and conclusions justified by the results?

No

Is the language acceptable?

Yes

Do you have any ethical concerns with this paper?

No

Have you any concerns about statistical analyses in this paper?

No

Recommendation?

Reject

Comments to the Author(s)

OVERALL

In this paper, the authors are extending their work on predictive tracking controllers (“Anticipation from sensation: using anticipating synchronization to stabilize a system with inherent sensory delay”) to applications in human-robot collaborative tasks. However, there are several shortcomings in their approach that make it difficult to see how the authors can meaningful transition this prior work to application. Moreover, the description of the controller is almost incomprehensible due to the lack of definition of the terms used.

GENERAL COMMENTS

The description of the controller is very hard to decipher. The block diagram presented in Figure 4 does not follow the typical notation. For instance, I assume that outputs of the “Plant” blocks in Figure 4 are q , not \dot{q} , but as written, the diagram conveys that \dot{q} is the output. I was able to rewrite the diagram and track the inputs and outputs, but I cannot definitively say that I know the input/output relationship of each block because the terms in the diagram of Figure 4 (1) do not match those found in the equations and (2) lack definitions. This section needs to be cleaned up.

The relevance of this control policy to human-robot physical collaboration is questionable. The fundamental challenge of human-robot collaboration is that the physical interaction is bidirectional. As the human acts upon the robot, the robot simultaneously acts on the human. In the model system tested, the simulated robot adapts to the simulated human, but the behavior of the simulated human is unaffected. The authors should be very explicit about this limitation and adjust their statements to downplay the relevance of this work human-robot physical interaction, unless it can be demonstrated otherwise.

Following the concern stated above, the authors describe the trajectory tracking task used in this study as being similar to “moving a piece of furniture, or operating one half of a saw” (e.g., page 4, lines 33-35). However, this is not quite true. In these tasks, a kinematic constraint is imposed. When two operators (be them humans or robots) are rigidly coupled through mutual object (also assumed to be rigid here), the distance between the operators’ endpoints is always fixed. Because of this kinematic coupling, errors in the robot tracking will impact the behavior of the human (i.e., the master). Again, this is not accounted for in the presented model system. Hence, the authors should refrain from referring to this as a “sawing” task, but rather a rhythmic trajectory tracking task.

The authors should put their work in the context of other prior research on human robot physical interaction during collaborative tasks (e.g., see work by Luka Peternel). Otherwise, it is hard to see how this advances the state of the art of human-robot collaboration.

The statement that the results of the simulated experiments can be “generalized to the physical robot” (page 4, line 3) needs to be substantiated. The authors state that the “robot is replaced with an accurate dynamical model” (page 3, line 56), but given that all models are wrong, the authors should further elaborate on what it is and is not included in the dynamic Baxter model (e.g., motor dynamics, noise, etc.). As stated in the paper, AS “relies on the fact that a deterministic dynamical system’s current state is strongly determined by its past” (page 1, lines 46-47). This begs the questions, would adding uncertainty in model or implementing this on hardware?

The advantage of using simulation over real hardware to test a wide range of coupling and delay terms is clear. However, the authors need to further demonstrate that their results in software translate onto real hardware. For instance, the authors should validate that the stable regions in parameter space observed in simulation are similar when implemented in hardware. And if they are not, the difference should be documented.

The choice to use a simulated Baxter robot as a testbed is perplexing given the fact that Baxter arm appears to be reduced to a 2 DOF planar robot. The authors should discuss their choice (or limited ability) to utilize all available degrees of freedom.

It is odd to me that the paper concludes with discussion of this work's "potential" relevance to exoskeleton control when this is not what was examined in the paper, nor was it mentioned anywhere else. It should be moved to the introduction or discussion. Again, the authors should keep in mind that there is a whole body of literature of describing exoskeleton control (even using delayed output feedback - e.g., Lim et al., (2019) Delayed Output Feedback Control for Gait Assistance with a Robotic Hip Exoskeleton) and human adaptation to exoskeletal devices. The authors should again put their proposed method in the context of this prior research.

Review form: Reviewer 2

Is the manuscript scientifically sound in its present form?

No

Are the interpretations and conclusions justified by the results?

No

Is the language acceptable?

Yes

Do you have any ethical concerns with this paper?

No

Have you any concerns about statistical analyses in this paper?

No

Recommendation?

Major revision is needed (please make suggestions in comments)

Comments to the Author(s)

This paper designs a controller to make a robot proactively collaborate with a human partner. This topic is important and the authors try to address a challenging problem of complicated master model. The reviewer appreciates the authors' attempt in this direction, but has several comments for the authors to address/explain:

- The reviewer could not understand if the internal model is for the robot or the master. From figure 4, it seems the it is for the robot, but then how can you predict the master's target when using the slave's model?

- Also in figure 4, why do you use the output of the internal model as the input of the controller u_{couple} ? Why not use the target directly?

- u_{couple} and u_{slave} only appear in figure 4, but are not explained in the main context.

What do they correspond to in the equations, e.g. u ?

- The problem is formulated by considering $d=0$, then it becomes a trajectory tracking problem. Does this formulation make sense and if yes, can we apply any controller for trajectory tracking in the literature, which has been extensively studied? What's the challenge of trajectory tracking for human-robot collaboration?

- The introduction needs to better reflect the state of the art in the field of Synchronisation-Based Control for a Collaborative Robot. Also other proactive controllers need to be compared in the experiment part, instead of just comparing with a PD controller.

- The system stability needs to be analysed in rigor instead of just explaining in text. This may also help the readers understand how the proposed method really works.

- The variables need to be explained and checked. For example, in equation 1 f is the master's dynamics but in 5 f is the slave's kinematics. For another example, how do you choose x_T in your experiment, in joint space?

The reviewer encourages a resubmission, but with clear explanations to address the above comments.

Review form: Reviewer 3

Is the manuscript scientifically sound in its present form?

Yes

Are the interpretations and conclusions justified by the results?

Yes

Is the language acceptable?

Yes

Do you have any ethical concerns with this paper?

No

Have you any concerns about statistical analyses in this paper?

No

Recommendation?

Accept with minor revision (please list in comments)

Comments to the Author(s)

This paper proposes a synchronization-based control paradigm for collaborative human-robot interactions. The method is interesting and of value to the community. Overall, the paper is well written and organized. However, several critical issues should be addressed before publication.

Major Issue:

1. Sect.2.3

I found it conceptually difficult to map Fig.4 to Eq. 3 and Eq.5-6. None of the symbols in the figure are used in the equations. I suggest the authors simplify the notation and be consistent across the entire manuscript (including the fig). For example, functions $f(\cdot)$ and $g(\cdot)$ in Fig.4 are not equivalent to those in Eq.1-2, however, a reader could be easily confused. What do the $C(\text{err})$ in the controller blocks represent? Where does Eq. 3 live in Fig. 4? In Eq. 3 what is symbol $v(t)$, velocity of the robot joints? How is Eq.3 a PD loop and is this K the same K in Fig.4? What is symbol B ?

2. Sect. 2.4

Can the author's elaborate on how accurate implementations of delay τ was accomplished, given the asynchronous nature of ROS? E.g., in Fig.8 can you guarantee that the delay time remained 0.01 (or 0.2) for the extend of the 30 s trial?

Some clarification is necessary in the following exert:

“This means that anticipation/lag is relative to the time the master signal is received by the slave, rather than its ‘transmission’, and the output of the system is compared to the input that caused it.”

When the authors refer to “anticipation/lag” is that the parameter τ ? Or the cross correlation reported in the results? What is “transmission” in this context? The output of what? And input of what?

The trouble is that the master, slave controller, and robot all have an outputs, and the slave controller, and robot have inputs. I’d recommend a careful rewrite of the above exert.

3. Sect. 2.5

Why is Eq.7 a PD controller?

The equation states that the velocity command is proportional to the error term with delayed feedback. There is no derivative term. Isn’t this just a proportional velocity controller?

4. Sect. 3

Additional clarification and rational is needed explain the use of the delayed feedback model as a comparison. Why do the authors compare AS to a controller which uses delayed feedback? In my understanding, AS uses delayed feedback in order to “anticipate” the reference signal, not because the signal itself is necessarily delayed (e.g., because of communication transmission, etc.). Thus, what is the value in comparing a delayed feedback control to AS?

In what scenario would robot pose be delayed? Why is this a “good” alternative to AS? If it is not, can you provide another model for comparison? (maybe PD controller without delay?)

In the introduction the authors state “while classical control becomes unstable and fails if the feedback from the plant is delayed..”, however this not an equal comparison to the AS approach since the delayed “feedback” in AS is actually beneficial and of a different nature.

Please clarify why/why not the comparison of AS to a delayed feedback system is warranted.

5. Generalizing the results on hardware would strength the paper. If this is not possible, please outline what are the necessary steps and challenges to realize the AS approach on hardware. E.g., sensing of the reference signal, physical coupling with human, etc.

Minor:

1. “group of ‘slave’ oscillators” in the abstract, is misleading, just use “slave oscillators”

2. “over the range of $0 < t < 0.2$ and $K > 5$.”

From Fig7.a when $K=13$ $0 < \tau < 0.2$ doesn’t seem smooth to me.

3. A reference frame in Fig. 3 would help conceptualize the coordinate system.

4. “This is preferable to using inverse kinematics because, in particular, the use of a Jacobian-based methods becomes difficult when information on the robot’s pose is delayed, as in the scenario of this experiment.” Isn’t the “delay” leveraged to anticipate the master? Presumably accurate Jacobians could be computed, while simultaneously storing “delay” copies for AS? Can the authors comment on this.

5. I would recommend succinctly outline the main contribution of this work in the final paragraph of the introduction. How is this work different than Ref.7? What is new?

Decision letter (RSOS-191762.R0)

19-Dec-2019

Dear Dr Eberle:

Manuscript ID RSOS-191762 entitled "Synchronisation-Based Control for a Collaborative Robot" which you submitted to Royal Society Open Science, has been reviewed. The comments from reviewers are included at the bottom of this letter.

In view of the criticisms of the reviewers, the manuscript has been rejected in its current form. However, a new manuscript may be submitted which takes into consideration these comments.

Please note that resubmitting your manuscript does not guarantee eventual acceptance, and that your resubmission will be subject to peer review before a decision is made.

Your resubmitted manuscript should be submitted by 17-Jun-2020. If you are unable to submit by this date please contact the Editorial Office.

on behalf of Dr Manoj Srinivasan (Associate Editor) and R. Kerry Rowe (Subject Editor)
openscience@royalsociety.org

Associate Editor Comments to Author (Dr Manoj Srinivasan):

While the reviewers see merit in the article, they have provided a number of suggestions for revisions and raised a few critical issues.

Reviewer 1 suggests substantially expanding the introduction to better place this article in the context of the available literature. Two of the reviewers felt that it was hard for them to understand how the control flow was structured and how the figures related to the equations. In general, more complete information regarding the technical aspects and the models used (e.g., the oscillator to be tracked) should be provided. It may be good to outline why the task selected is a worthwhile goal in collaborative robotics, what the specific application context is: show how the proposed anticipatory tracking algorithm may perform when the human reacts to the robot motion as well, as reviewer 1 implicitly suggests.

Reviewers' Comments to Author:

Reviewer: 1

Comments to the Author(s)

OVERALL

In this paper, the authors are extending their work on predictive tracking controllers ("Anticipation from sensation: using anticipating synchronization to stabilize a system with inherent sensory delay") to applications in human-robot collaborative tasks. However, there are several shortcomings in their approach that make it difficult to see how the authors can meaningful transition this prior work to application. Moreover, the description of the controller is almost incomprehensible due to the lack of definition of the terms used.

GENERAL COMMENTS

The description of the controller is very hard to decipher. The block diagram presented in Figure 4 does not follow the typical notation. For instance, I assume that outputs of the "Plant" blocks in Figure 4 are q , not \dot{q} , but as written, the diagram conveys that \dot{q} is the output. I was able to rewrite the diagram and track the inputs and outputs, but I cannot definitively say that I know the input/output relationship of each block because the terms in the diagram of Figure 4 (1) do not match those found in the equations and (2) lack definitions. This section needs to be cleaned up.

The relevance of this control policy to human-robot physical collaboration is questionable. The fundamental challenge of human-robot collaboration is that the physical interaction is bidirectional. As the human acts upon the robot, the robot simultaneously acts on the human. In the model system tested, the simulated robot adapts to the simulated human, but the behavior of the simulated human is unaffected. The authors should be very explicit about this limitation and adjust their statements to downplay the relevance of this work human-robot physical interaction, unless it can be demonstrated otherwise.

Following the concern stated above, the authors describe the trajectory tracking task used in this study as being similar to "moving a piece of furniture, or operating one half of a saw" (e.g., page 4, lines 33-35). However, this is not quite true. In these tasks, a kinematic constraint is imposed. When two operators (be them humans or robots) are rigidly coupled through mutual object (also assumed to be rigid here), the distance between the operators' endpoints is always fixed. Because of this kinematic coupling, errors in the robot tracking will impact the behavior of the human (i.e., the master). Again, this is not accounted for in the presented model system. Hence, the authors should refrain from referring to this as a "sawing" task, but rather a rhythmic trajectory tracking task.

The authors should put their work in the context of other prior research on human robot physical interaction during collaborative tasks (e.g., see work by Luka Peternel). Otherwise, it is hard to see how this advances the state of the art of human-robot collaboration.

The statement that the results of the simulated experiments can be "generalized to the physical robot" (page 4, line 3) needs to be substantiated. The authors state that the "robot is replaced with an accurate dynamical model" (page 3, line 56), but given that all models are wrong, the authors should further elaborate on what it is and is not included in the dynamic Baxter model (e.g., motor dynamics, noise, etc.). As stated in the paper, AS "relies on the fact that a deterministic dynamical system's current state is strongly determined by its past" (page 1, lines 46-47). This begs the questions, would adding uncertainty in model or implementing this on hardware?

The advantage of using simulation over real hardware to test a wide range of coupling and delay

terms is clear. However, the authors need to further demonstrate that their results in software translate onto real hardware. For instance, the authors should validate that the stable regions in parameter space observed in simulation are similar when implemented in hardware. And if they are not, the difference should be documented.

The choice to use a simulated Baxter robot as a testbed is perplexing given the fact that Baxter arm appears to be reduced to a 2 DOF planar robot. The authors should discuss their choice (or limited ability) to utilize all available degrees of freedom.

It is odd to me that the paper concludes with discussion of this work's "potential" relevance to exoskeleton control when this is not what was examined in the paper, nor was it mentioned anywhere else. It should be moved to the introduction or discussion. Again, the authors should keep in mind that there is a whole body of literature of describing exoskeleton control (even using delayed output feedback – e.g., Lim et al., (2019) Delayed Output Feedback Control for Gait Assistance with a Robotic Hip Exoskeleton) and human adaptation to exoskeletal devices. The authors should again put their proposed method in the context of this prior research.

Reviewer: 2

Comments to the Author(s)

This paper designs a controller to make a robot proactively collaborate with a human partner. This topic is important and the authors try to address a challenging problem of complicated master model. The reviewer appreciates the authors' attempt in this direction, but has several comments for the authors to address/explain:

- The reviewer could not understand if the internal model is for the robot or the master. From figure 4, it seems the it is for the robot, but then how can you predict the master's target when using the slave's model?

- Also in figure 4, why do you use the output of the internal model as the input of the controller u_{couple} ? Why not use the target directly?

- u_{couple} and u_{slave} only appear in figure 4, but are not explained in the main context.

What do they correspond to in the equations, e.g. u ?

- The problem is formulated by considering $d=0$, then it becomes a trajectory tracking problem. Does this formulation make sense and if yes, can we apply any controller for trajectory tracking in the literature, which has been extensively studied? What's the challenge of trajectory tracking for human-robot collaboration?

- The introduction needs to better reflect the state of the art in the field of Synchronisation-Based Control for a Collaborative Robot. Also other proactive controllers need to be compared in the experiment part, instead of just comparing with a PD controller.

- The system stability needs to be analysed in rigor instead of just explaining in text. This may also help the readers understand how the proposed method really works.

- The variables need to be explained and checked. For example, in equation 1 f is the master's dynamics but in 5 f is the slave's kinematics. For another example, how do you choose x_T in your experiment, in joint space?

The reviewer encourages a resubmission, but with clear explanations to address the above comments.

Reviewer: 3

Comments to the Author(s)

This paper proposes a synchronization-based control paradigm for collaborative human-robot interactions. The method is interesting and of value to the community. Overall, the paper is well written and organized. However, several critical issues should be addressed before publication.

Major Issue:

1. Sect.2.3

I found it conceptually difficult to map Fig.4 to Eq. 3 and Eq.5-6. None of the symbols in the figure are used in the equations. I suggest the authors simplify the notation and be consistent across the entire manuscript (including the fig). For example, functions $f(\cdot)$ and $g(\cdot)$ in Fig.4 are not equivalent to those in Eq.1-2, however, a reader could be easily confused. What do the $C(\text{err})$ in the controller blocks represent? Where does Eq. 3 live in Fig. 4? In Eq. 3 what is symbol $v(t)$, velocity of the robot joints? How is Eq.3 a PD loop and is this K the same K in Fig.4? What is symbol B ?

2. Sect. 2.4

Can the author's elaborate on how accurate implementations of delay τ was accomplished, given the asynchronous nature of ROS? E.g., in Fig.8 can you guarantee that the delay time remained 0.01 (or 0.2) for the extend of the 30 s trial?

Some clarification is necessary in the following exert:

"This means that anticipation/lag is relative to the time the master signal is received by the slave, rather than its 'transmission', and the output of the system is compared to the input that caused it."

When the authors refer to "anticipation/lag" is that the parameter τ ? Or the cross correlation reported in the results? What is "transmission" in this context? The output of what? And input of what?

The trouble is that the master, slave controller, and robot all have an outputs, and the slave controller, and robot have inputs. I'd recommend a careful rewrite of the above exert.

3. Sect. 2.5

Why is Eq.7 a PD controller?

The equation states that the velocity command is proportional to the error term with delayed feedback. There is no derivative term. Isn't this just a proportional velocity controller?

4. Sect. 3

Additional clarification and rational is needed explain the use of the delayed feedback model as a comparison. Why do the authors compare AS to a controller which uses delayed feedback? In my understanding, AS uses delayed feedback in order to "anticipate" the reference signal, not because the signal itself is necessarily delayed (e.g., because of communication transmission, etc.). Thus, what is the value in comparing a delayed feedback control to AS?

In what scenario would robot pose be delayed? Why is this a "good" alternative to AS? If it is not, can you provide another model for comparison? (maybe PD controller without delay?)

In the introduction the authors state "while classical control becomes unstable and fails if the feedback from the plant is delayed...", however this not an equal comparison to the AS approach since the delayed "feedback" in AS is actually beneficial and of a different nature.

Please clarify why/why not the comparison of AS to a delayed feedback system is warranted.

5. Generalizing the results on hardware would strength the paper. If this is not possible, please

outline what are the necessary steps and challenges to realize the AS approach on hardware. E.g., sensing of the reference signal, physical coupling with human, etc.

Minor:

1. "group of 'slave' oscillators" in the abstract, is misleading, just use "slave oscillators"
2. "over the range of $0 < t < 0.2$ and $K > 5$." From Fig7.a when $K=13$ $0 < \tau < 0.2$ doesn't seem smooth to me.
3. A reference frame in Fig. 3 would help conceptualize the coordinate system.
4. "This is preferable to using inverse kinematics because, in particular, the use of a Jacobian-based methods becomes difficult when information on the robot's pose is delayed, as in the scenario of this experiment." Isn't the "delay" leveraged to anticipate the master? Presumably accurate Jacobians could be computed, while simultaneously storing "delay" copies for AS? Can the authors comment on this.
5. I would recommend succinctly outline the main contribution of this work in the final paragraph of the introduction. How is this work different than Ref.7? What is new?

Author's Response to Decision Letter for (RSOS-191762.R0)

See Appendix A.

RSOS-201267.R0

Review form: Reviewer 2

Is the manuscript scientifically sound in its present form?

No

Are the interpretations and conclusions justified by the results?

No

Is the language acceptable?

Yes

Do you have any ethical concerns with this paper?

No

Have you any concerns about statistical analyses in this paper?

No

Recommendation?

Major revision is needed (please make suggestions in comments)

Comments to the Author(s)

The reviewer appreciates the authors' effort to address some of my comments in the first round of review. Especially the authors have clearly discussed the importance of prediction of the master's movement to make the slave's action proactive. However, there are still some issues that need to be further addressed:

- The major issue is how the internal models (5) and (6) provide prediction capabilities is not clear. The prediction is only shown by simulation results but why it works is not explained, e.g. using theoretical analysis.
- This work claims contributions to the field of human-robot collaboration, but the simulations do not include any kind of 'human'. Therefore, whether the internal models can be used to predict human movements is questionable.
- Figure 8 is a wrong figure, as it's not related to the caption.
- What is u in equation 5?

Review form: Reviewer 3

Is the manuscript scientifically sound in its present form?

Yes

Are the interpretations and conclusions justified by the results?

Yes

Is the language acceptable?

Yes

Do you have any ethical concerns with this paper?

No

Have you any concerns about statistical analyses in this paper?

No

Recommendation?

Accept as is

Comments to the Author(s)

The authors have adequately addressed my original concerns.

Decision letter (RSOS-201267.R0)

Dear Dr Eberle

The Editors assigned to your paper RSOS-201267 "Synchronisation-Based Control for a Collaborative Robot" have now received comments from reviewers and would like you to revise the paper in accordance with the reviewer comments and any comments from the Editors. Please note this decision does not guarantee eventual acceptance.

Please submit your revised manuscript and required files (see below) no later than 21 days from today's (ie 22-Oct-2020) date. Note: the ScholarOne system will 'lock' if submission of the revision is attempted 21 or more days after the deadline. If you do not think you will be able to meet this deadline please contact the editorial office immediately.

on behalf of Dr Manoj Srinivasan (Associate Editor) and R. Kerry Rowe (Subject Editor)
openscience@royalsociety.org

Associate Editor Comments to Author (Dr Manoj Srinivasan):

The author has mostly addressed the reviewer concerns. One reviewer has additional concerns, which I encourage the authors to address, by providing additional clarifications, fixing the typos/omissions, providing appropriate caveats, and/or appropriately weakening the claims made.

I have another suggestion/concern with the use of the traditional robotic terminology, specifically, the "master and slave system". See, for instance, concerns below:
[https://en.wikipedia.org/wiki/Master/slave_\(technology\)#Terminology_concerns](https://en.wikipedia.org/wiki/Master/slave_(technology)#Terminology_concerns)
I would recommend that you consider either modifying the terminology or consider acknowledging the problematic aspects of the terminology in the article.

Reviewer comments to Author:

Reviewer: 2
Comments to the Author(s)

The reviewer appreciates the authors' effort to address some of my comments in the first round of review. Especially the authors have clearly discussed the importance of prediction of the master's movement to make the slave's action proactive. However, there are still some issues that need to be further addressed:

- The major issue is how the internal models (5) and (6) provide prediction capabilities is not clear. The prediction is only shown by simulation results but why it works is not explained, e.g. using theoretical analysis.
- This work claims contributions to the field of human-robot collaboration, but the simulations do not include any kind of 'human'. Therefore, whether the internal models can be used to predict human movements is questionable.
- Figure 8 is a wrong figure, as it's not related to the caption.
- What is u in equation 5?

Reviewer: 3

Comments to the Author(s)

The authors have adequately addressed my original concerns.

===PREPARING YOUR MANUSCRIPT===

- one version identifying all the changes that have been made (for instance, in coloured highlight, in bold text, or tracked changes);a 'clean' version of the new manuscript that incorporates the changes made, but does not highlight them. This version will be used for typesetting if your manuscript is accepted. Please ensure that any equations included in the paper are editable text and not embedded images.

===PREPARING YOUR REVISION IN SCHOLARONE===

Please ensure that you include a summary of your paper at Step 2 'Type, Title, & Abstract'. This should be no more than 100 words to explain to a non-scientific audience the key findings of your

research. This will be included in a weekly highlights email circulated by the Royal Society press office to national UK, international, and scientific news outlets to promote your work.

<https://royalsociety.org/journals/authors/author-guidelines/#supplementary-material> to include a suitable title and informative caption. An example of appropriate titling and captioning may be found at https://figshare.com/articles/Table_S2_from_Is_there_a_trade-off_between_peak_performance_and_performance_breadth_across_temperatures_for_aerobic_scops_in_teleost_fishes_/3843624.

Author's Response to Decision Letter for (RSOS-201267.R0)

See Appendix B.

Decision letter (RSOS-201267.R1)

Dear Dr Eberle,

It is a pleasure to accept your manuscript entitled "Synchronisation-Based Control for a Collaborative Robot" in its current form for publication in Royal Society Open Science.

At this stage, we ask that you please archive your GitHub code within the Zenodo repository: <https://guides.github.com/activities/citable-code/>. By doing this, a formal, citable DOI will be associated with your data record, and an open license (CC-BY preferred) can be applied to your data. We would then ask that you please update your data availability statement to read as:

"Data and relevant code for this research work are stored in GitHub: [GitHub URL here] and have been archived within the Zenodo repository: <https://doi.org/zenodo.....> [ref number].

on behalf of Dr Manoj Srinivasan (Associate Editor) and R. Kerry Rowe (Subject Editor)
openscience@royalsociety.org

Appendix A

Response to Reviewers

We would like to thank all the reviewers for the time they spent on our manuscript and the many good points they raised. The following document contains our responses to these comments, as well as detailing changes we have made to the paper that we hope will alleviate their concerns.

Since we have made a large number of changes to the manuscript, we have attached a highlighted version which indicates where we have made changes.

Questions from reviewers are in **bold**, responses are in plain text, and quoted text from the manuscript is in *italics*.

Reviewer 1

Overview

In this paper, the authors are extending their work on predictive tracking controllers (“Anticipation from sensation: using anticipating synchronization to stabilize a system with inherent sensory delay”) to applications in human-robot collaborative tasks. However, there are several shortcomings in their approach that make it difficult to see how the authors can meaningful transition this prior work to application. Moreover, the description of the controller is almost incomprehensible due to the lack of definition of the terms used.

1. The description of the controller is very hard to decipher. The block diagram presented in Figure 4 does not follow the typical notation. For instance, I assume that outputs of the “Plant” blocks in Figure 4 are q , not q_dot , but as written, the diagram conveys that q_dot is the output. I was able to rewrite the diagram and track the inputs and outputs, but I cannot definitively say that I know the input/output relationship of each block because the terms in the diagram of Figure 4 (1) do not match those found in the equations and (2) lack definitions. This section needs to be cleaned up.

We have redesigned the block diagram to be more visually readable, and made sure its symbols and terms are consistent with the text. [Page 6]

Figure 4. Block diagram of the proposed control scheme, where the velocity command v to the plant (Baxter) is a linear combination of a velocity control signal $B\dot{q}_{slave}$ produced by an internal control loop with specific ‘slave’ dynamics, and a second ‘coupling’ control signal $u_{couple} = KM[x_{slave}(t) - x_{plant}(t - \tau)]$ based on delayed feedback from the robot itself, where B and K are scalar constants and M is a transformation matrix. The slave model $h(q_{slave}, x_{err})$ tracks the position of the target x_{target} by minimising the difference x_{err} between x_{target} and the model’s ‘end effector’ calculated using a copy of Baxter’s forward kinematics ($x_{slave} = s(q_{slave})$). The internal control loop imposes a specific dynamical behaviour on the plant, causing it to act as an AS slave that can anticipate the target’s movements. Thus, the addition of the coupling term causes the robot to lead the target, if the feedback delay (τ) is large enough. (Green blocks represent intrinsic elements of the underlying system, while yellow blocks have been added to enable anticipation.)

2. The relevance of this control policy to human-robot physical collaboration is questionable. The fundamental challenge of human-robot collaboration is that the physical interaction is bidirectional. As the human acts upon the robot, the robot simultaneously acts on the human. In the model system tested, the simulated robot adapts to the simulated human, but the behavior of the simulated human is unaffected. The authors should be very explicit about this limitation and adjust their statements to downplay the relevance of this work human-robot physical interaction, unless it can be demonstrated otherwise.

In human-human interaction the physical interaction constitutes a form of tacit communication because both agents can reason about the other's intentions. Rather than attempt to simulate this communication, which would require a specialised platform to sense the human's movements, we aimed to make the robot's behaviour as predictable as possible. Hence, a control framework where the robot behaves in the same way as if it is reacting to the user, but with a much smaller latency than would be possible without anticipation. Within the limited context of the chosen task the correct response is always to maintain constant distance from the partner, and as long as the anticipation is robust to varied input, it can be considered successful. We have clarified that this study pertains to this specific circumstance, and not the wider problem of human-robot cooperation.

"The experimental can be tailored to the intrinsic timescale and capabilities of the Baxter platform: a back-and-forth coordinated motion in which the robot must follow a moving target representing its partner without lagging or becoming unstable. This simulates the timing aspect of bimanual handling (e.g. moving one end of a piece of furniture) without mutual physical interaction that would require a specialised platform to investigate." [Page 3, paragraph 5]

3. Following the concern stated above, the authors describe the trajectory tracking task used in this study as being similar to "moving a piece of furniture, or operating one half of a saw" (e.g., page 4, lines 33-35). However, this is not quite true. In these tasks, a kinematic constraint is imposed. When two operators (be them humans or robots) are rigidly coupled through mutual object (also assumed to be rigid here), the distance between the operators' endpoints is always fixed. Because of this kinematic coupling, errors in the robot tracking will impact the behavior of the human (i.e., the master). Again, this is not accounted for in the presented model system. Hence, the authors should refrain from referring to this as a "sawing" task, but rather a rhythmic trajectory tracking task.

We agree that this is a reasonable criticism, and have replaced all references to 'sawing'.

4. The authors should put their work in the context of other prior research on human robot physical interaction during collaborative tasks (e.g., see work by Luka Peternel). Otherwise, it is hard to see how this advances the state of the art of human-robot collaboration.

We have added additional citations to the introduction and included more explanation of how we believe this work fits into the context of the field.

"A common approach is to attempt to infer the human partner's intentions from their behaviour so that the robot can 'preempt' their future actions. However, while many stereotypical human movements are well characterised (such as simple reaching motions, by the minimum-jerk model [1]), there is no accepted complete model of human behaviour. Various methods have been used to

work around this limitation, such as in work by Thobbi et al. where the robot switches to a reactive follower role when its prediction of the human's motion diverges from its sensor measurements [2]. Mainprice et al. attempted to learn the objective function of human reaching movements through inverse reinforcement learning, taking the assumption that humans operate on the principle of optimal control [3], while Bussy et al. decomposed their collaborative task into motor primitives with rigidly-defined trajectories [4]" **Page 1, paragraph 3]**

5. The statement that the results of the simulated experiments can be “generalized to the physical robot” (page 4, line 3) needs to be substantiated. The authors state that the “robot is replaced with an accurate dynamical model” (page 3, line 56), but given that all models are wrong, the authors should further elaborate on what it is and is not included in the dynamic Baxter model (e.g., motor dynamics, noise, etc.). As stated in the paper, AS “relies on the fact that a deterministic dynamical system’s current state is strongly determined by its past” (page 1, lines 46-47). This begs the questions, would adding uncertainty in model or implementing this on hardware?

The simulator encompasses the robot’s kinematics, dynamics and the real-time control loop that implements its safety constraints. Given that the simulator was designed by the robot’s manufacturers for the express purpose of allowing users to develop control schemes that would generalise onto the physical robot, we believe this is a reasonable assumption. If the uncertainty in the model was large enough that its controllability was significantly reduced this would have an effect. The model does not encompass noise), but neither did the robot exhibit a level of noise large enough for this to represent a qualitative difference.

“During the testing phase, in order to evaluate the control under an appropriate range of conditions within a reasonable time, and without risking wear-and-tear to the robot’s components, the motion of the robot can be simulated by an accurate dynamical model, simulated in Gazebo [18] (see Fig.2). This replicates all of Baxter’s software interfaces as well as its characteristic dynamical behaviour, allowing the results of the simulated experiments to be generalised to the physical robot [22]" **[Page 4, paragraph 2]**

6. The advantage of using simulation over real hardware to test a wide range of coupling and delay terms is clear. However, the authors need to further demonstrate that their results in software translate onto real hardware. For instance, the authors should validate that the stable regions in parameter space observed in simulation are similar when implemented in hardware. And if they are not, the difference should be documented.

Due to the limited availability of the robot, the system was prototyped in hardware, then moved into simulation for in-depth testing. The differences between hardware and software are a valid concern, but we are confident that the simulation encompasses all the salient features that significantly influence the results. We hope that our clarifications about the simulation in our other responses address this concern.

7. The choice to use a simulated Baxter robot as a testbed is perplexing given the fact that Baxter arm appears to be reduced to a 2 DOF planar robot. The authors should discuss their choice (or limited ability) to utilize all available degrees of freedom.

The primary reason for using the Baxter robot was that its series elastic actuators gave it a more complex dynamical behaviour than a standard manipulator arm. While our preference would have been for a simpler robot with these properties, at the time we did not have the option of building or acquiring one, and making use of the additional degrees of freedom added little to the experiment (in our opinion).

“Rather than creating a special-purpose robot that could call into question the broader applicability of this study, the Baxter collaborative robot was chosen as a testing model. Baxter’s hardware and software safety features provide a reasonable estimate of the constraints which a collaborative control rule can be expected to have, and it has a high-quality simulator that can be used for repeated experiments [16]. The experimental can be tailored to the intrinsic timescale and capabilities of the Baxter platform: a back-and-forth coordinated motion in which the robot must follow a moving target representing its partner without lagging or becoming unstable. This simulates the timing aspect of bimanual handling (e.g. moving one end of a piece of furniture) without mutual physical interaction that would require a specialised platform to investigate.” [Page 2, paragraph 5]

8. It is odd to me that the paper concludes with discussion of this work’s “potential” relevance to exoskeleton control when this is not what was examined in the paper, nor was it mentioned anywhere else. It should be moved to the introduction or discussion. Again, the authors should keep in mind that there is a whole body of literature of describing exoskeleton control (even using delayed output feedback – e.g., Lim et al., (2019) Delayed Output Feedback Control for Gait Assistance with a Robotic Hip Exoskeleton) and human adaptation to exoskeletal devices. The authors should again put their proposed method in the context of this prior research.

In light of the changes made due to the other comments in this review, we felt it more sense to remove this theme from the conclusion.

Reviewer 2:

Overall

This paper designs a controller to make a robot proactively collaborate with a human partner. This topic is important and the authors try to address a challenging problem of complicated master model. The reviewer appreciates the authors' attempt in this direction, but has several comments for the authors to address/explain:

1. The reviewer could not understand if the internal model is for the robot or the master. From figure 4, it seems the it is for the robot, but then how can you predict the master's target when using the slave's model?

The internal model does not strictly reproduce the target (master) or the robot (slave), but is designed to exhibit the dynamics of a system that can anticipate a master with continuous motion. We have altered the text throughout the manuscript to make this clearer.

2. Also in figure 4, why do you use the output of the internal model as the input of the controller u_{couple} ? Why not use the target directly?

This is a quirk of the specific ROS implementation we used and plays no functional role vs. using the target directly. It occurred to us afterwards that the result was aesthetically unpleasing when put in block diagram form, but changing it in good faith would require rerunning the experiment for only a small improvement in performance.

3. u_{couple} and u_{slave} only appear in figure 4, but are not explained in the main context. What do they correspond to in the equations, e.g. u ?

We have standardised the equation terminology throughout the paper, including figures.

4. The problem is formulated by considering $d=0$, then it becomes a trajectory tracking problem. Does this formulation make sense and if yes, can we apply any controller for trajectory tracking in the literature, which has been extensively studied? What's the challenge of trajectory tracking for human-robot collaboration?

The specific challenge in trajectory tracking for human-robot collaboration (as we see it) is not accuracy or stability, which are well-addressed problems, but responsiveness. It is not a given that robots that interact closely with humans will be capable of the high motor gains required to track their human partner without some form of prediction (in fact, this may be undesirable for safety reasons). Therefore, we consider it desirable to have a way to anticipate a partner with unknown motor behaviour, as systematically predicting a human's intentions is an enormous practical and theoretical burden on the controller.

Since referencing a constant displacement d and then setting it to 0 could be viewed as redundant, we have changed the text in section 2.2 to make it clear that the robot is specifically matching the motion of the target. **[Section 2.2: Collaborative task]**

5. The introduction needs to better reflect the state of the art in the field of Synchronisation-Based Control for a Collaborative Robot. Also other proactive controllers need to be compared in the experiment part, instead of just comparing with a PD controller.

We have rewritten the introduction, including additional citations explaining to position this study relative to the state of the art. In terms of the experimental procedure, we have followed a similar approach to Thobbi et al. [1], where a novel proactive controller is compared to its reactive analogue. Given the large number of approaches to predicting human motion, I believe this is the fairest way to present the results, with less room for unconscious cherry-picking. [Section 1: Introduction]

6. The system stability needs to be analysed in rigor instead of just explaining in text. This may also help the readers understand how the proposed method really works.

It is a typical practice in works on anticipating synchronisation to determine the stability bounds numerically, and we would expect a full analytical characterisation of the stability to require a second, larger study. We have added a section discussing our method for analysing stability and placing it in the context of previous AS research.

“As explained by Voss [8], the stability of an AS system of the form in Eq.2 is dependent on the transversal system $\Delta = x(t) - y(t - \tau)$ having a stable fixed point at $\Delta = 0$ (the anticipating manifold). For pairs of identical linear systems the stability conditions for the manifold can be determined by calculating its complex Lyapunov exponent [9], yielding an explicit dependence on delay τ and coupling strength K . Such an analytical solution does not exist for the nonlinear, nonidentical case, but the stability of the manifold with respect to τ and K can be determined numerically by constructing a phase diagram. By systematically varying τ and K over a series of simulations, it is possible to visualise the stability of the anticipation manifold in the $K - \tau$ plane. As in [9], we use the correlation coefficient and correlation lag to identify the region in the $K - \tau$ plane in which the anticipation manifold is stable, as stable regions share the characteristics of negative lag (the slave leads the master) and high correlation coefficient (the slave closely matches the master). A simplified visualisation of one of these phase diagrams is given in Fig. 7, showing a characteristic division between stable and unstable regions. In the results below the phase diagrams are presented as heatmaps in order to show the variation in prediction time within the stable region.” [Page 9, paragraph 2]

Figure 7. Region (white) in which the anticipation manifold is stable, bounded by the curve $K\tau = 6$ and the line $K = 5$.

7. The variables need to be explained and checked. For example, in equation 1 f is the master's dynamics but in 5 f is the slave's kinematics. For another example, how do you choose x_T in your experiment, in joint space?

We have edited the equations to change ambiguous and overlapping symbols. x_T is a vector of the midpoints for each joints range of motion, representing the 'resting point' for the relaxation and oscillator dynamics. We have added text to make this clear.

" x_{T} is a vector of 'resting points' representing the centre of each joint's range of motion." [Page 7, paragraph 2]

Reviewer: 3

Overview

This paper proposes a synchronization-based control paradigm for collaborative human-robot interactions. The method is interesting and of value to the community. Overall, the paper is well written and organized. However, several critical issues should be addressed before publication.

Major Issues:

1. Sect.2.3: I found it conceptually difficult to map Fig.4 to Eq. 3 and Eq.5-6. None of the symbols in the figure are used in the equations. I suggest the authors simplify the notation and be consistent across the entire manuscript (including the fig). For example, functions $f(\cdot)$ and $g(\cdot)$ in Fig.4 are not equivalent to those in Eq.1-2, however, a reader could be easily confused. What do the $C(\text{err})$ in the controller blocks represent? Where does Eq. 3 live in Fig. 4? In Eq. 3 what is symbol $v(t)$, velocity of the robot joints? How is Eq.3 a PD loop and is this K the same K in Fig.4? What is symbol B ?

We have standardised the symbols used in equations throughout the paper and added additional explanation where new symbols are introduced.

2. Sect. 2.4: Can the author's elaborate on how accurate implementations of delay τ was accomplished, given the asynchronous nature of ROS? E.g., in Fig.8 can you guarantee that the delay time remained 0.01 (or 0.2) for the extend of the 30 s trial?

It is true that ROS's asynchronous nature makes tracking delay times problematic. For this reason, a buffer was implemented *within* the slave node that delayed each input signal for a fixed number of (fixed length) iterations. Furthermore, the node published each of its outputs paired with the corresponding input for that iteration. In this way, the delay interval remained constant with respect to the slave system, which is what the results reflect. We have clarified this in the text.

"Because of ROS's asynchronous implementation the exact relationship between the timings of the master and slave nodes cannot be guaranteed. For this reason a buffer was implemented within the slave node that delays and stores the feedback from the robot for a number of (0.01s) iterations. For each iteration the node publishes the master, slave, and robot feedback values used to update the slave's state, which are recorded as experimental data. This means that the additional feedback delay remains consistent with respect to the slave system for any given time step, and the output of the system is compared to the input that caused it." [Page 8, paragraph 3]

2(a): Some clarification is necessary in the following exert:

"This means that anticipation/lag is relative to the time the master signal is received by the slave, rather than its 'transmission', and the output of the system is compared to the input that caused it."

When the authors refer to "anticipation/lag" is that the parameter τ ? Or the cross correlation reported in the results? What is "transmission" in this context? The output of what? And input of

what?

The trouble is that the master, slave controller, and robot all have an outputs, and the slave controller, and robot have inputs. I'd recommend a careful rewrite of the above exert.

We believe the response to the previous comment also applies here.

3. Sect. 2.5: Why is Eq.7 a PD controller?

The equation states that the velocity command is proportional to the error term with delayed feedback. There is no derivative term. Isn't this just a proportional velocity controller?

You are correct, there were some elements of an older version of the paper that carried through unnoticed – we have amended the description.

“Both candidate slave systems are tested over a range of feedback delay ($\tau=0.01-0.6s$) and coupling strength ($K=0-30$) and compared against the performance of a basic proportional controller that uses the same transformation matrix (Eq.7).” [Page 8, paragraph 4]

4. Sect. 3: Additional clarification and rational is needed explain the use of the delayed feedback model as a comparison. Why do the authors compare AS to a controller which uses delayed feedback? In my understanding, AS uses delayed feedback in order to “anticipate” the reference signal, not because the signal itself is necessarily delayed (e.g., because of communication transmission, etc.). Thus, what is the value in comparing a delayed feedback control to AS?

In what scenario would robot pose be delayed? Why is this a “good” alternative to AS? If it is not, can you provide another model for comparison? (maybe PD controller without delay?)

The intention of this controller is to make delayed feedback from the plant equivalent to the delayed feedback term in AS. Establishing the link between anticipation and delay in the feedback term for this controller additionally proves that delayed feedback due to environmental factors will also cause anticipation. The proportional controller used for comparison was operating in the minimum delay condition of our test platform, but to be clear the effect does not disappear when feedback delay is removed, as there is an intrinsic delay inherent in the normal feedback loop. Using an explicit feedback delay term was considered the best way to make the phenomenon obvious, though lowering the controller gain would have a similar effect.

5. In the introduction the authors state “while classical control becomes unstable and fails if the feedback from the plant is delayed...”, however this not an equal comparison to the AS approach since the delayed “feedback” in AS is actually beneficial and of a different nature.

Please clarify why/why not the comparison of AS to a delayed feedback system is warranted.

The delay in AS is introduced intentionally to produce a beneficial effect, but it is only beneficial due to the design of the system, not any quality of the delay itself. The aim of this work is to design a feedback controller that responds similarly to delay to an AS system without compromising the ‘goal’ of the control task.

5. Generalizing the results on hardware would strength the paper. If this is not possible, please outline what are the necessary steps and challenges to realize the AS approach on hardware. E.g., sensing of the reference signal, physical coupling with human, etc.

The work this paper is based on was begun in hardware (hence the use of a simulated Baxter robot, rather than a generic model), however unavoidable life events meant it had to be finished with minimal access to the physical robot. We have made changes that reference the difficulties inherent in physical coupling with a human.

“Rather than creating a special-purpose robot that could call into question the broader applicability of this study, the Baxter collaborative robot was chosen as the testing model. Baxter’s hardware and software safety features provide a reasonable estimate of the constraints which a collaborative control rule can be expected to have to work within, and it has a high-quality simulator that can be used for repeated experiments. The experimental task was tailored to the intrinsic timescale and capabilities of the Baxter platform: a back-and-forth coordinated motion in which the robot must follow a moving target representing its partner without lagging or becoming unstable. This simulates the timing aspect of bimanual handling (e.g. moving one end of a piece of furniture) without mutual physical interaction that would require a specialised platform to investigate. Over the course of the experiment the feedback delay between robot and partner was varied, along with the coupling term K , to establish that this behaviour is robust. In order to observe the full parameter range in a reasonable time, the experiment was performed using an accurate Gazebo simulation of the Baxter robot” [Page 2, paragraph 5]

Minor issues:

1. “group of ‘slave’ oscillators” in the abstract, is misleading, just use “slave oscillators”

We have revised this phrasing. [Page 1, paragraph 1]

2. “over the range of $0 < t < 0.2$ and $K > 5$.”

From Fig7.a when $K=13$ $0 < \tau < 0.2$ doesn’t seem smooth to me.

Smooth may be a mischaracterisation: we have removed this phrasing.

“in Fig.7a, it can be seen that the lead time (represented by positive values) increases over the range of $0 < \tau < 0.2$ and $K > 5$.” [Page 9, paragraph 1]

3. A reference frame in Fig. 3 would help conceptualize the coordinate system.

This is a very reasonable request, and we have updated the figure and caption to provide more context to what it represents.

Figure 3. Illustration of simulated manual coordination task: the robot (green, follower) is controlled by the AS control system and must match the movements of the leader (red) as closely as possible, despite its pseudorandom motion.

4. “This is preferable to using inverse kinematics because, in particular, the use of a Jacobian-based methods becomes difficult when information on the robot’s pose is delayed, as in the scenario of this experiment.” Isn’t the “delay” leveraged to anticipate the master? Presumably accurate Jacobians could be computed, while simultaneously storing “delay” copies for AS? Can the authors comment on this.

This is an interesting question. Since a Jacobian-based transformation is an instantaneous solution to the inverse problem, the difference between two sets of coordinates transformed by two Jacobians calculated at different time points would raise the question of what the term represents kinematically. Since this would require further discussion that is not strictly relevant to the main contribution, we decided to use a controller with simpler geometric constraints.

“Because s_1 also effects the radial coordinate, M is not a true coordinate transformation, however since the joints are effectively coupled through the forward kinematics, e_1 can continuously compensate for s_1 ’s effects on r . This limits the arm to variations on a single pose, allowing the static linear transformation to ensure convergence on the target. This is preferable to using inverse kinematics because, in particular, the use of a Jacobian-based methods becomes problematic when calculating the difference between two different poses separated by a delay, as in the coupling term. [Page 6, paragraph 3]

5. I would recommend succinctly outline the main contribution of this work in the final paragraph of the introduction. How is this work different than Ref.7? What is new?

We have added such a paragraph.

“This study aims to demonstrate that a control system designed from the bottom up to take advantage of AS can enable a robot to anticipate the motion of an autonomous partner, even without an accurate model. By tying the velocities of its joints to a series of simulated dynamical systems, the robot's own delayed sensory feedback becomes the key element in an AS coupling that drives it to follow its partner with minimal (or even slightly negative) latency. This represents a categorically different approach from methods of collaborative control based on explicit models, which decline in performance if the partner is not correctly characterised.” [Page 2, paragraph 6]

References:

1. Thobbi, A., Gu, Y., Sheng, W. Using human motion estimation for human-robot cooperative manipulation. 2011 IEEE/RSJ International Conference on Intelligent Robots and Systems 2873–2878 (2011).

Appendix B

Response to Reviewers

We would like to thank you for taking the time to read our initial revisions – please find an itemised list of our changes below. We hope you find them to your satisfaction.

1. **I have another suggestion/concern with the use of the traditional robotic terminology, specifically, the "master and slave system". See, for instance, concerns below: [https://en.wikipedia.org/wiki/Master/slave_\(technology\)#Terminology_concerns](https://en.wikipedia.org/wiki/Master/slave_(technology)#Terminology_concerns) I would recommend that you consider either modifying the terminology or consider acknowledging the problematic aspects of the terminology in the article.**

All instances of the terms “master” and “slave” have been replaced by “leader” and “follower”, respectively.

2. **The major issue is how the internal models (5) and (6) provide prediction capabilities is not clear. The prediction is only shown by simulation results but why it works is not explained, e.g. using theoretical analysis.**

A pair of additional paragraphs has been added to explain how the prediction capability arises.

Page 6, paragraph 2/Page 7, paragraph 1: *“The follower’s prediction occurs via a simple variation on dynamical synchronisation. While an ordinary synchronisation coupling ($[x(t)-y(t)]$) represents a difference term that drives the follower y to evolve towards the current state of the leader x ($[x(t)=y(t)]$), a delay coupling ($[x(t)-y(t-\tau)]$) approximates a term that accelerates the follower’s time evolution towards the leader’s future state ($[x(t+\tau)=y(t)]$), assuming they are sufficiently similar. Work by Hayashi et al. [9] revealed that this is due to the renormalisation of time in the follower: when $y(t-\tau)$ from Eq.2 is Taylor-expanded it gives the new timestep $t^*=t/(1-k\tau)$, causing the follower to evolve faster than the leader until it ‘catches up’ with its future state. Provided a transversal system $\Delta=f_1(x)-f_2(y)-K\Delta$ can be defined (where $\Delta=x-y$) that has a fixed point at $\Delta=0$ (which is a trivial solution), there exists some $\tau<\tau_0$ for which that fixed point is locally attracting [10]. The size of this region cannot be guaranteed, and can only be estimated numerically for any pair of systems, which motivated the selection of follower dynamics that were established from prior research to have a significant anticipation period for more complex leaders. In this study, even though the autonomous dynamics of the follower system were replaced with leaky integrators and harmonic oscillators, the parallel structure of the control scheme (Fig. 4) reinforces this internal model to induce resonance with the dynamics of the leader system, resulting in the creation of similar dynamics..”*

3. **This work claims contributions to the field of human-robot collaboration, but the simulations do not include any kind of 'human'. Therefore, whether the internal models can be used to predict human movements is questionable.**

We have weakened this claim appropriately throughout the document: human manual tracking experiments are still referenced as part of the inspiration for the experiment, but the paper now avoids drawing conclusions related to human partners. In a future study, it would be interesting to have a human participant interacting with the Baster based on the

proposed A-S control. As we tested the dynamics of the leader, not fully rhythmic nor chaotic, our tested method can be a milestone towards the Human-robot collaboration.

In addition, we have modified figure 3 to show a more abstract partner, rather than a clearly human hand:

4. Figure 8 is a wrong figure, as it's not related to the caption

Our apologies for this mistake. This has been addressed – please see the matched figure and caption below.

Figure 8. Cross-correlation peak lag (a) and peak correlation coefficient (b) plotted against coupling strength K and delay τ , where the dynamics of the internal model are governed by Eq.5. Where $K > 5$, and barring the unstable region bounded by $K\tau = 6$ where the correlation coefficient is close to 0, the robot leads the target, as indicated by positive cross-correlation lag values.

5. What is u in equation 5?

This was a remnant of the previous equation terminology, and has been brought in line with the rest of the paper.